# The study of pentagonal chain with respect to schultz index, modified schultz index, schultz polynomial and modified schultz polynomial

**Guofeng Yu[1], Shahid Zaman[2], Mah Jabeen[2], Xuewu Zuo[3]***

**1** Public Courses Education Department, Anhui Business Vocational College, Hefei, Anhui, China,
**2** Department of Mathematics, University of Sialkot, Sialkot, Pakistan, **3** General Education Department, Anhui Xinhua University, Hefei, China

\* xinhuazxw@163.com

## Abstract

Distance-based topological indices are numerical parameters that are derived from the distances between atoms in a molecular structure, and they provide a quantitative measure of the topology and geometry of a molecule. The distance-based topological indices uses to predict various properties of molecules, including their boiling points, melting points, and solubility. It also predicts the biological activity of molecules, including their pharmacological and toxicological properties. Pentagonal chain molecules are organic compounds that consist of a linear chain of five-membered (pentagons) connected by carbon and bonds. These molecules have unique structural and electronic properties that make them useful in a variety of applications. Motivated by the pentagonal chain molecules, we have considered a pentagonal chain graph and it is denoted by $P_n$. We have computed some distance based topological indices for $P_n$. The paper focuses on a pentagonal chain molecules denoted by G, and derives several distance-based topological indices. These indices compromise insights into physicochemical properties, aid in identifying structural characterizations, and enhance understanding of molecular properties.

**Data Availability Statement:** All relevant data are within the manuscript.

## 1. Introduction

Let $G = (V_G, E_G)$ be the graph, where $V_G$ is the set of vertices, and $E_G$ is the set of edges. The size of a graph is determined by the number of edges, and the distance between two vertices is determined by the shortest path and denoted by $d(x,y)$.

Topological indices (TIs) play a vital role in mathematical chemistry. There are two fundamental types of graph TIs: degree-based and distance-based TIs. In this paper, we used the class of distance-based TIs for the pentagonal chain graph. Distance is an important factor in graph theory, since it effects the graph's structure and algebraic properties while also generating a variety of critical distance-based parameters such as normal distance, diameter, radius, eccentricity, distance matrix, and resistance distance.

A mathematical technique used to any graph that models a molecule structure is called a TI graph, also known as a molecular description. In quantitative structure-property relationship

**Funding:** The author(s) received no specific funding for this work.

**Competing interests:** The authors have declared that no competing interests exist.

and quantitative structure-activity relationship studies, molecular descriptors play a basic role in mathematical chemistry. The Wiener index [1] which is based on the distance between pairs of vertices or points in each network, is the most widely used TI. In a simple connected undirected graph G, the Wiener index (WI) of the graph is sum of the distances over all pairs of vertices. The graph G corresponds to what is known as the molecular graph of an organic compound. The study of a sharp lower bound for WI(G) of an random graph in terms of the order, size and diameter is given in [2]. An upper bound on the Wiener index of a k-connected graph is discussed in [3]. The Wiener index [4] is one of the important topological index uses to analyze the central properties of molecule structure in chemistry. Mostly, Wiener index is useful in a molecular graph to examine the configuration of organic molecules like Cyclo alkane and related research areas. The Wiener index is defined mathematically as,

$$W(P_n) = \sum_{x,y \subseteq V(P_n)} d(x, y)$$

The Gutman index (GI) has been considered in previous works, such as [5] and [6, 7]. The author named as Feng incorporated the Gutman index for unicycle graphs and related to polycyclic particles in [7]. Additionally, Feng [7] also studied bicyclic graphs. Gutman said that the GI is expected value to the WI, as noted in [6]. The formula for a Gutman index is as follows:

$$Gut(P_n) = \sum_{x,y \subseteq V(P_n)} dx.dy.d(x, y)$$

The Schultz index [8] presented by H. P. Schultz in 1989 and defined as

$$S_c(P_n) = \frac{1}{2} \sum_{x,y \subseteq V(P_n)} (dx + dy).d(x, y)$$

is basically a weighted version of WI, also proposed another weighted version of Schultz index [9] which is used to established some properties of Schultz molecular topological index. The Schultz index can also express in terms of the Wiener index, by multiplying the degree sum with the distances in a chemical graph. Some interesting results on Schultz index are computed in [10, 11] and [12]. The uses of distance based descriptors are briefly examined in quantitative structural–physical relationship are studied in [13]. The other distance based topological indices are characterized in [14–20] and the resistance distances and eigenvalues based topological indices are obtained in [21–28].

A comparison of the Schultz molecular topological index with the Wiener index is given in [29]. Inspired from Schultz index the modified Schultz index is introduced in [30]. Some results on modified schultz index and Wiener index of some graphs are discussed in [31, 32]. The modified Schultz index of the graph G is mathematical form as:

$$S_c^*(P_n) = \frac{1}{2} \sum_{x,y \subseteq V(P_n)} (dx.dy).d(x, y)$$

The multiplicative Zagreb index for k-trees are discussed in [33], the polynomials version of Schultz, Modified Schultz, and Hosoya are investigated in [34] and [35].The Schultz polynomial and its modified form are utilized as molecular descriptors and are described as

$$S_c(P_n, \alpha) = \frac{1}{2} \sum_{x,y \subseteq V(P_n)} (dx + dy)\alpha^{d(x,y)}$$

and,

$$S_c^*(P_n, \alpha) = \frac{1}{2} \sum_{x,y \subseteq V(P_n)} (dx.dy) \alpha^{d(x,y)}$$

These distance based structural descriptors and their polynomials, are mostly scheduled and figured in [34]. Some other results are discussed in [36, 37]. The Schultz, Modified Schultz polynomials and their topological indices of Jahangir graphs are obtained in [38]. Recently, Ullah et al., obtained some interesting results on structural characterizations of graphs through degree based topological indices [39–43] and Mondal et al., studied the molecular graphs in [44–47]. On the other hand, some other properties of topological indices are discussed by Ahmad et al., [48–51] and Zaman et al., in [52–59]. The authors He et al., [60] defined the pentagonal chain graph, as a simple graph consisting of five vertices arranged in each cycle.

## 2. The structural analysis of pentagonal chain graph

Let $P_n$ be a pentagonal chain with n pentagons as shown in Fig 1. In theoretical chemistry, molecules are frequently represented as graphs, with atoms serving as vertices and chemical bonds as edges. The pentagonal chain graph is a basic model for depicting linear chemical chains or segments within larger compounds. Numerous results have been done in hexagonal systems, because of its very significant applications in theoretical chemistry. The hexagonal systems are natural graph representations of benzenoid hydrocarbon and they have been of great interest and extensively studied [61, 62]. Motivated from the hexagonal systems, we have considered the pentagonal chain network and computed some distance based topological indices.

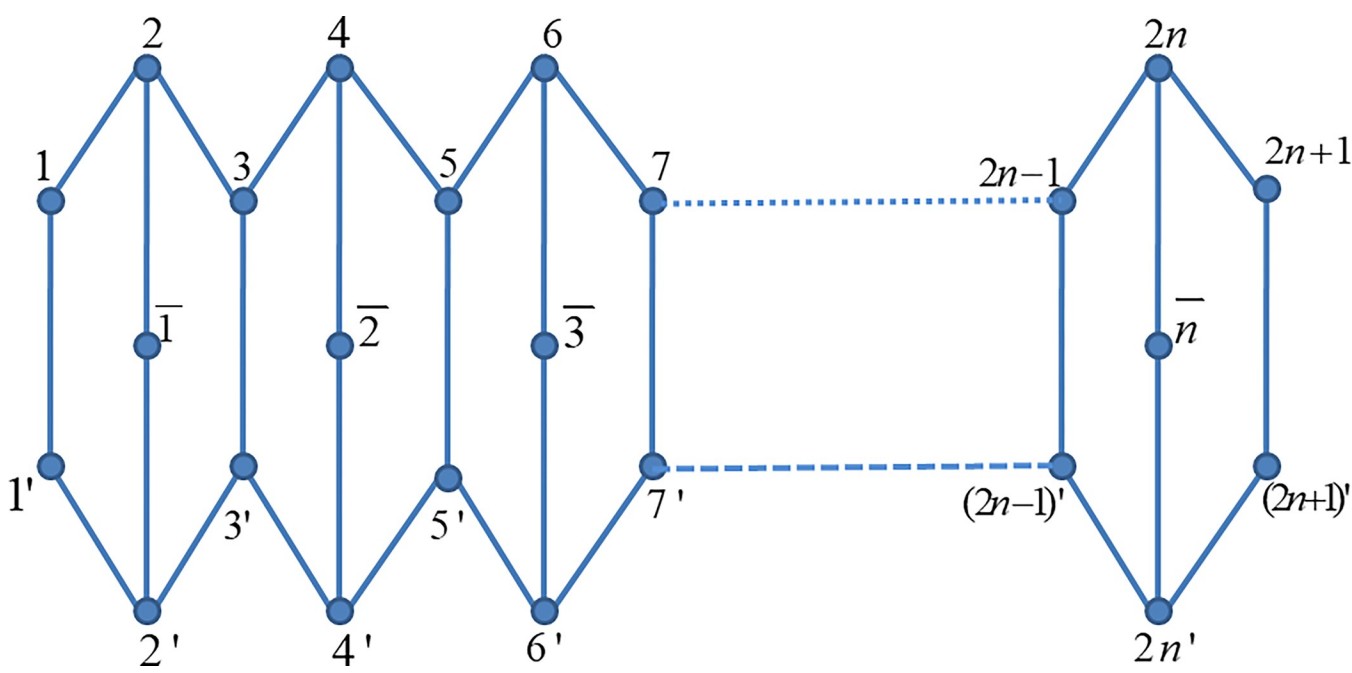

**Fig 1. The pentagonal chain graph $P_n$.**

## 3. Main results

The graph shown in Fig 1 is a pentagonal chain graph, and several important indices have been stated in this section via a distance-based method. Following are the computational findings:

### 3.1. Theorem

Let $P_n$ be a pentagonal chain graph, then the Wiener index of $P_n$ is as:

$$W(P_n) = \frac{25n^3 + 57n^2 + 23n + 3}{3}$$

**Proof:** Firstly, we calculate the distances $d_{x,y}$ for all points (fixed $x$ and $y$). Since, the vertices of $P_n$ are given in Fig 1. Then the expression of each type of points are:

- vertex 1 of $P_n$:

$$f(n) = \left[\sum_{\vartheta=1}^{2n-1} \vartheta + \sum_{\vartheta=1}^{2n-1}(\vartheta+1)\right] + \left[2n+1 + (2n+1) + \sum_{\vartheta=1}^{\omega} 2\vartheta\right]$$

$$= \left[2\sum_{\vartheta=1}^{2n-1} \vartheta + \sum_{\vartheta=1}^{2n-1} 1\right] + \left[4n + 2(1) + 2\sum_{\vartheta=1}^{n} \vartheta\right]$$

$$= \left(\frac{2(2n-1)(2n)}{2} + 2n - 1\right) + \left(4n + 2 + \frac{2n(n+1)}{2}\right)$$

$$= 5n^2 + 5n + 1$$

- vertex $2\varphi(1 \leq \varphi \leq n)$ of $P_n$:

$$f(2\varphi, n) = \left(\sum_{\vartheta=1}^{\varphi}(2\vartheta-1) + \sum_{\vartheta=1}^{n-\varphi}(2\vartheta+1) + (4n+2)\right) + \left(2 + \sum_{\vartheta=1}^{2\varphi-2}(2\vartheta+1) + \sum_{\vartheta=1}^{2n-2\varphi}(2\vartheta+1)\right)$$

$$= \left(\sum_{\vartheta=1}^{\varphi} 2\vartheta - \sum_{\vartheta=1}^{\varphi} 1 + \sum_{\vartheta=1}^{n-\varphi} 2\vartheta + \sum_{\vartheta=1}^{n-\varphi} 1 + (4n+2)\right) + \left(2 + \sum_{\vartheta=1}^{2\varphi-2} 2\vartheta + \sum_{\vartheta=1}^{2\varphi-2} 1 + \sum_{\vartheta=1}^{2n-2\varphi} 2\vartheta + \sum_{\vartheta=1}^{2n-2\varphi} 1\right)$$

$$= \left(\frac{2\varphi(\varphi+1)}{2} - \varphi + \frac{2(n-\varphi)(n-\varphi+1)}{2} + n - \varphi + 4n + 2\right)$$
$$+ \left(2 + \frac{2(2\varphi-2)(2\varphi-1)}{2} + 2\varphi - 2 + \frac{2(2n-2\varphi)(2n-2\varphi+1)}{2} + 2n - 2\varphi\right)$$

$$= 5n^2 - 10n\varphi - 10\varphi + 10\varphi^2 + 10n + 4$$

- vertex $2\varphi{+}1(1 \leq \varphi \leq n{-}1)$ of $P_\omega$;

$$f(2\varphi+1, n) = \left(\sum_{\vartheta=1}^{\varphi} 2\vartheta + \sum_{\vartheta=1}^{n-\varphi} 2\vartheta + (4n+2)\right) + \left(1 + \sum_{\vartheta=1}^{2\varphi-1}(2\vartheta+1) + \sum_{\vartheta=1}^{2n-2\varphi-1}(2\vartheta+1)\right)$$

$$= (2\sum_{\vartheta=1}^{\varphi} \vartheta + 2\sum_{\vartheta=1}^{n-\varphi} \vartheta + 2(2n+1)) + (1 + 2\sum_{\vartheta=1}^{2\varphi-1} \vartheta + \sum_{\vartheta=1}^{2\varphi-1} 1 + 2\sum_{\vartheta=1}^{2n-2\varphi-1} \vartheta + \sum_{\vartheta=1}^{2n-2\varphi-1} 1)$$

$$= \left(\frac{2\varphi(\varphi+1)}{2} + \frac{2(n-\varphi)(n-\varphi+1)}{2} + 4n + 2\right)$$
$$+ \left(1 + \frac{2(2\varphi-1)(2\varphi)}{2} + 2\varphi - 1 + \frac{2(2n-2\varphi-1)(2n-2\varphi)}{2} + 2n - 2\varphi - 1\right)$$

$$= 5n^2 - 10n\varphi + 10\varphi^2 + 5n + 1$$

vertex $\bar{\varphi}(1 \le \bar{\varphi} \le n)$ of $P_\omega$:

$$f(\bar{\varphi},\omega) = 2(\sum_{\vartheta=1}^{\varphi-1}(\vartheta+1) + \sum_{\vartheta=1}^{n-\varphi}(\vartheta+1) + (2n+2)) + 2(1 + \sum_{\vartheta=1}^{2\varphi-2}(\vartheta+1) + \sum_{\vartheta=1}^{2n-2\varphi}(\vartheta+1))$$

$$= 2(\sum_{\vartheta=1}^{\varphi-1} \vartheta + \sum_{\vartheta=1}^{\varphi-1} 1 + \sum_{\vartheta=1}^{n-\varphi} \vartheta + \sum_{\vartheta=1}^{n-\varphi} 1 + 2n + 2(1)) + 2(1 + \sum_{\vartheta=1}^{2\varphi-2} \vartheta + \sum_{\vartheta=1}^{2\varphi-2} 1 + \sum_{\vartheta=1}^{2n-2\varphi} \vartheta + \sum_{\vartheta=1}^{2n-2\varphi} 1)$$

$$= 2\left(\frac{\varphi(\varphi-1)}{2} + \varphi - 1 + \frac{(n-\varphi)(n-\varphi+1)}{2} + n - \varphi + 2n + 2\right)$$
$$+ 2\left(1 + \frac{(2\varphi-2)(2\varphi-1)}{2} + 2\varphi - 2 + \frac{(2n-2\varphi+1)(2n-2\varphi)}{2} + 2n - 2\varphi\right)$$

$$= 5n^2 - 10n\varphi + 10\varphi^2 - 10\varphi + 13n + 2$$

By using the formula of Wiener index one has

$$W(P_n) = \frac{4f(n) + 2\sum_{\varphi=1}^{n} f(2\varphi, n) + 2\sum_{\varphi=1}^{n-1} f(2\varphi+1, n) + \sum_{\varphi=1}^{n} f(\bar{\varphi}, n)}{2}$$

$$W(P_n) = \frac{25n^3 + 57n^2 + 23n + 3}{3}$$

## 3.2. Theorem

Let $P_n$ be the pentagonal chain graph with pentagons, then the Gutman index of pentagonal chain graph is as:

$$Gut(P_n) = \frac{196n^3 + 276n^2 + 71n + 3}{3}$$

**Proof:** Here we have partitioned the vertex set into 4 parts, then put the results in the general formula of Gutman index

■ vertex 1 of $P_n$;

$$f(n) = 2.3\left[\sum_{\vartheta=1}^{2n-1} \vartheta + \sum_{\vartheta=1}^{2n-1}(\vartheta+1)\right] + 2.2\left[2n + 1 + (2n+1) + \sum_{\vartheta=1}^{n} 2\vartheta\right]$$

$$= 2.3 \left[ 2 \sum_{\vartheta=1}^{2n-1} \vartheta + \sum_{\vartheta=1}^{2n-1} 1 \right] + 2.2 \left[ 4n + 2(1) + 2 \sum_{\vartheta=1}^{n} \vartheta \right]$$

$$= 2.3 \left[ \frac{2(2n-1)2n}{2} + 2n - 1 \right] + 2.2 \left[ 4n + 2 + \frac{2n(n+1)}{2} \right]$$

$$= 28n^2 + 20n + 2$$

- vertex $2\varphi(1 \leq \varphi \leq n)$ of $\mathrm{P}_n$;

$$f(2\varphi, n) = 3.2 \left[ \sum_{\vartheta=1}^{\varphi} (2\vartheta - 1) + \sum_{\vartheta=1}^{n-\varphi} (2\vartheta + 1) + (4n + 2) \right]$$

$$+ 3.3 \left[ 2 + \sum_{\vartheta=1}^{2\varphi-2} (2\vartheta + 1) + \sum_{\vartheta=1}^{2n-2\varphi} (2\vartheta + 1) \right]$$

$$= 6 \left[ 2(2n+1) + \sum_{\vartheta=1}^{\varphi} (2\vartheta - 1) + \sum_{\vartheta=1}^{n-\varphi} (2\vartheta + 1) \right] + 9 \left[ 2 + \sum_{\vartheta=1}^{2\varphi-2} (2\vartheta + 1) + \sum_{\vartheta=1}^{2n-2\varphi} (2\vartheta + 1) \right]$$

$$= 6 \left[ 4n + 2 + \frac{2\varphi(\varphi+1)}{2} - \varphi + \frac{2(n-\varphi)(n-\varphi+1)}{2} + n - \varphi \right]$$
$$+ 3.3 \left[ 2 + \frac{2(2\varphi-2)(2\varphi-1)}{2} + 2\varphi - 2 + \frac{2(2n-2\varphi)(2n-2\varphi+1)}{2} + 2n - 2\varphi \right]$$

$$= 42n^2 - 84n\varphi + 84\varphi^2 - 84\varphi + 72n + 30$$

- vertex $2\varphi+1(1 \leq \varphi \leq n-1)$ of $\mathrm{P}_n$;

$$f(2\varphi+1, n) = 3.2 \left[ \sum_{\vartheta=1}^{\varphi} 2\vartheta + \sum_{\vartheta=1}^{n-\varphi} 2\vartheta + (4n+2) \right] + 3.3 \left[ 1 + \sum_{\vartheta=1}^{2\varphi-1} (2\vartheta + 1) + \sum_{\vartheta=1}^{2n-2\varphi-1} (2\vartheta + 1) \right]$$

$$= 6 \left[ 2 \sum_{\vartheta=1}^{\varphi} \vartheta + 2 \sum_{\vartheta=1}^{n-\varphi} \vartheta + 2(2n+1) \right] + 9 \left[ 1 + \sum_{\vartheta=1}^{2\varphi-1} (2\vartheta + 1) + \sum_{\vartheta=1}^{2n-2\varphi-1} (2\vartheta + 1) \right]$$

$$= 6 \left[ \frac{2\varphi(\varphi+1)}{2} + \frac{2(n-\varphi)(n-\varphi+1)}{2} + 4n + 2 \right]$$
$$+ 9 \left[ 1 + \frac{2(2\varphi-1)(2\varphi)}{2} + 2\varphi - 1 + \frac{2(2n-2\varphi-1)(2n-2\varphi)}{2} + 2n - 2\varphi - 1 \right]$$

$$= 42n^2 - 84n\varphi + 84\varphi^2 + 30n + 3$$

vertex $\bar{\varphi}\,(1 \leq \bar{\varphi} \leq n)$ of $P_n$;

$$f(\bar{\varphi}, n) = 2.2.2\left[2n + 2 + \sum_{\vartheta=1}^{\varphi-1}(\vartheta+1) + \sum_{\vartheta=1}^{n-\varphi}(\vartheta+1)\right] + 2.2.3\left[1 + \sum_{\vartheta=1}^{2\varphi-2}(\vartheta+1) + \sum_{\vartheta=1}^{2n-2\varphi}(\vartheta+1)\right]$$

$$= 2.2.2\left[2(n+1) + \sum_{\vartheta=1}^{\varphi-1}\vartheta + \sum_{\vartheta=1}^{\varphi-1}1 + \sum_{\vartheta=1}^{n-\varphi}\vartheta + \sum_{\vartheta=1}^{n-\varphi}1\right] + 2.2.3\left[1 + \sum_{\vartheta=1}^{2\varphi-2}\vartheta + \sum_{\vartheta=1}^{2\varphi-2}1 + \sum_{\vartheta=1}^{2n-2\varphi}\vartheta + \sum_{\vartheta=1}^{2n-2\varphi}1\right]$$

$$= 8\left(2n + 2 + \frac{\varphi(\varphi-1)}{2} + \varphi - 1 + \frac{(n-\varphi)(n-\varphi+1)}{2} + n - \varphi\right)$$

$$+ 12\left(1 + \frac{(2\varphi-2)(2\varphi-1)}{2} + 2\varphi - 2 + \frac{(2n-2\varphi)(2n-2\varphi+1)}{2} + 2n - 2\varphi\right)$$

$$= 28n^2 - 56n\varphi + 56\varphi^2 - 56\varphi + 64n + 8$$

By summing all above one has

$$Gut(P_n) = \frac{4f(n) + 2\sum_{\varphi=1}^{n}f(2\varphi, n) + 2\sum_{\varphi=1}^{n-1}f(2\varphi+1, n) + \sum_{\varphi=1}^{n}f(\bar{\varphi}, n)}{2}$$

$$= \frac{196n^3 + 276n^2 + 71n + 3}{3}$$

Hence proved.

### 3.3. Theorem

Let $P_n$ denotes the pentagonal chain graph, then the schultz index of $P_n$ is given as

$$S_c(P_n) = \frac{70n^3 + 129n^2 + 41n + 3}{3}$$

**Proof:** According to proof of theorem 3.2 we computed the schultz index of $P_n$.
■ vertex 1 of $P_n$:

$$f(n) = \frac{1}{2}\left[(2+3)\left\{\sum_{\vartheta=1}^{2n-1}\vartheta + \sum_{\vartheta=1}^{2n-1}(\vartheta+1)\right\} + (2+2)\left\{2n + 1 + (2n+1) + \sum_{\vartheta=1}^{n}2\vartheta\right\}\right]$$

$$= \frac{1}{2}\left\{10\sum_{\vartheta=1}^{2n-1}\vartheta + 5\sum_{\vartheta=1}^{2n-1}1 + 4(2(2n+1) + 2\sum_{\vartheta=1}^{n}\vartheta)\right\}$$

$$= \frac{1}{2}\left\{\frac{10(2n-1)(2n)}{2} + 5(2n-1) + 16n + 8 + \frac{8n(n+1)}{2}\right\}$$

$$= \frac{1}{2}(24n^2 + 20n + 3)$$

$$= 12n^2 + 10n + \frac{3}{2}$$

- vertex $2\varphi(1\leq\varphi\leq n)$ of $P_n$;

$$f(2\varphi, n) = \frac{1}{2}[(3+2)\left[\sum_{\vartheta=1}^{\varphi}(2\vartheta-1) + \sum_{\vartheta=1}^{n-\varphi}(2\vartheta+1) + (4n+2)\right] + (3+3)\left[2 + \sum_{\vartheta=1}^{2\varphi-2}(2\vartheta+1) + \sum_{\vartheta=1}^{2n-2\varphi}(2\vartheta+1)\right]]$$

$$= \frac{1}{2}[(3+2)\left[2\sum_{\vartheta=1}^{\varphi}\vartheta - \sum_{\vartheta=1}^{\varphi}1 + 2\sum_{\vartheta=1}^{n-\varphi}\vartheta + \sum_{\vartheta=1}^{n-\varphi}1 + 2(2n+1)\right] + (3+3)\left[2 + 2\sum_{\vartheta=1}^{2\varphi-2}\vartheta + \sum_{\vartheta=1}^{2\varphi-2}1 + 2\sum_{\vartheta=1}^{2n-2\varphi}\vartheta + \sum_{\vartheta=1}^{2n-2\varphi}1\right]]$$

$$= \frac{1}{2}[5\left(\frac{2\varphi(\varphi+1)}{2} - \varphi + \frac{2(n-\varphi)(n-\varphi+1)}{2} + n - \varphi + 4n + 2\right)$$

$$+ 6\left(2 + \frac{2(2\varphi-2)(2\varphi-1)}{2} + 2\varphi - 2 + \frac{2(2n-2\varphi)(2n-2\varphi+1)}{2} + 2n - 2\varphi\right)]$$

$$= \frac{1}{2}[5(n^2 - 2n\varphi - 2\varphi + 2\varphi^2 + 6n + 2) + 6(4n^2 - 8n\varphi - 8\varphi + 8\varphi^2 + 4n + 2)]$$

$$= \frac{1}{2}[29n^2 + 58\varphi^2 - 58n\varphi - 58\varphi + 54n + 22]$$

- vertex $2\varphi+1(1\leq\varphi\leq n-1)$ of $P_n$;

$$f(2\varphi+1, n) = \frac{1}{2}\left[\begin{array}{l}(2+3)\{\sum_{\vartheta=1}^{\varphi}2\vartheta + \sum_{\vartheta=1}^{n-\varphi}2\vartheta + (4n+2)\} \\ +(3+3)\{1 + \sum_{\vartheta=1}^{2\varphi-1}(2\vartheta+1) + \sum_{\vartheta=1}^{2n-2\varphi-1}(2\vartheta+1)\}\end{array}\right]$$

$$= \frac{1}{2}\left[(5)\{\sum_{\vartheta=1}^{\varphi}2\vartheta + \sum_{\vartheta=1}^{n-\varphi}2\vartheta + 2(2n+1)\} + (6)\{1 + 2\sum_{\vartheta=1}^{2\varphi-1}\vartheta + \sum_{\vartheta=1}^{2\varphi-1}1 + 2\sum_{\vartheta=1}^{2n-2\varphi-1}\vartheta + \sum_{\vartheta=1}^{2n-2\varphi-1}1\}\right]$$

$$= 5\frac{2\varphi(\varphi+1)}{2} + \frac{2(n-\varphi)(n-\varphi+1)}{2} + 4n + 2\} + 61 + \frac{2(2\varphi-1)(2\varphi)}{2} + 2\varphi - 1$$

$$+ \frac{2(2n-2\varphi-1)(2n-2\varphi)}{2} + 2n - 2\varphi - 1\}$$

$$= \frac{1}{2}[5(n^2 + 2\varphi^2 - 2n\varphi + 5n + 2) + 6(4n^2 + 8\varphi^2 - 8n\varphi - 1)]$$

$$= \frac{1}{2}[29n^2 + 58\varphi^2 - 58n\varphi + 25n + 4]$$

vertex $\bar{\varphi}(1 \leq \bar{\varphi} \leq n)$ of $P_n$;

$$f(\bar{\varphi}, n) = \frac{1}{2}\left[2.(2+2)\{\sum_{\vartheta=1}^{\varphi-1}(\vartheta+1) + \sum_{\vartheta=1}^{n-\varphi}(\vartheta+1) + (2n+2)\} + 2.(2+3)\{1 + \sum_{\vartheta=1}^{2\varphi-2}(\vartheta+1) + \sum_{\vartheta=1}^{2n-2\varphi}(\vartheta+1)\}\right]$$

$$= \frac{1}{2}\left[(8)\{\sum_{\vartheta=1}^{\varphi-1}\vartheta + \sum_{\vartheta=1}^{\varphi-1}1 + \sum_{\vartheta=1}^{n-\varphi}\vartheta + \sum_{\vartheta=1}^{n-\varphi}1 + (2n+2)\} + (10)\{1 + \sum_{\vartheta=1}^{2\varphi-2}\vartheta + \sum_{\vartheta=1}^{2\varphi-2}1 + \sum_{\vartheta=1}^{n-\varphi}\vartheta + \sum_{\vartheta=1}^{2n-2\varphi}1\}\right]$$

$$= \frac{1}{2} \left[ \begin{array}{l} 8\frac{\varphi(\varphi-1)}{2} + \varphi - 1 + \frac{(n-\varphi)(n-\varphi+1)}{2} + n - \varphi + 2(n+1) \\ +101 + \frac{(2\varphi-1)(2\varphi-2)}{2} + 2\varphi - 2 + \frac{(2n-2\varphi)(2n-2\varphi+1)}{2} + 2n - 2\varphi \end{array} \right]$$

$$= \frac{1}{2} [4(n^2 + 2\varphi^2 - 2n\varphi - \varphi + 7n + 2) + 10(2n^2 + 4\varphi^2 - 4n\varphi - 4\varphi + 3n)]$$

$$= 12n^2 + 24\varphi^2 - 24n\varphi - 24\varphi + 29n + 4$$

Sum of all and then divided by two,

$$S_c(\mathrm{P}_n) = \frac{4f(n) + 2\sum_{\varphi=1}^{n} f(2\varphi, n) + 2\sum_{\varphi=1}^{n-1} f(2\varphi+1, n) + \sum_{\varphi=1}^{n} f(\bar{\varphi}, n)}{2}$$

$$S_c(\mathrm{P}_n) = \frac{70n^3 + 129n^2 + 41n + 3}{3}$$

Hence proved,

### 3.4. Theorem

Assume that $\mathrm{P}_n$ is pentagonal chain then modified schultz index is given as

$$S_c^*(\mathrm{P}_n) = \frac{1}{6}(196n^3 + 150n^2 + 197n + 3)$$

**Proof:** From theorem 3.2 we obtain the following four vertex partitions

$$\blacksquare\ f_1^*(n) = 14n^2 + 10n + 1$$

$$\blacksquare\ f_2^*(2\varphi, n) = 21n^2 + 42\varphi^2 - 42n\varphi - 42\varphi + 36n + 15$$

$$\blacksquare\ f_3^*(2\varphi+1, n) = 21n^2 + 42\varphi^2 - 42n\varphi + 15n + \frac{3}{2}$$

$$\blacksquare\ f_4^*(\bar{\varphi}, n) = 14n^2 - 28n\varphi - 28\varphi + 28\varphi^2 + 32n + 4$$

Based on the general formula of modified schultz index one has

$$S_c^*(\mathrm{P}_n) = \frac{4f(n) + 2\sum_{\varphi=1}^{n} f(2\varphi, n) + 2\sum_{\varphi=1}^{n-1} f(2\varphi+1, n) + \sum_{\varphi=1}^{n} f(\bar{\varphi}, n)}{2}$$

$$S_c^*(\mathrm{P}_n) = \frac{1}{6}(196n^3 + 150n^2 + 197n + 3)$$

Hence, we obtained the desired result.

### 3.5. Theorem

Let $P_n$ be a pentagonal chain graph then the Schultz polynomial index is as

$$S_c(P_n, \alpha) = \frac{1}{2} \sum_{x,y \subseteq V(P_n)} (dx + dy).\alpha^{d(x,y)}$$

**Proof:** Since, we have already calculated the Schultz index, so based on those results we have divided the vertices into four sets as

■ vertex 1 of $P_n$;

$$f(n) = \frac{1}{2} \left[ (2+3)\{ \sum_{\vartheta=1}^{2n-1} \alpha^\vartheta + \sum_{\vartheta=1}^{2n-1} \alpha^{\vartheta+1} \} + (2+2)\{ \alpha + \sum_{\vartheta=1}^{n} \alpha^{2\vartheta} + \alpha^{2n} + \alpha^{2n+1} \} \right]$$

$$= \frac{1}{2} \left[ (2+3)\left\{ \frac{\alpha^{2n} - \alpha}{\alpha - 1} + \frac{\alpha(\alpha^{2n} - \alpha)}{\alpha - 1} \right\} + (2+2)\left\{ \alpha + \alpha^{2n} + \alpha^{2n+1} + \frac{\alpha^2(\alpha^{2n} - 1)}{\alpha^2 - 1} \right\} \right]$$

$$= \frac{1}{2} \left[ 5\left( \frac{\alpha^{2n+1} + \alpha^{2n} - \alpha^2 - \alpha}{\alpha - 1} \right) + 6\left( \alpha + \alpha^{2n} + \alpha^{2n+1} + \frac{\alpha^2(\alpha^{2n} - 1)}{\alpha^2 - 1} \right) \right]$$

$$= \frac{4\alpha^{2n+3} + 13\alpha^{2n+2} + 6\alpha^{2n+1} - \alpha^{2n} - \alpha^3 - 14\alpha^2 - 9\alpha}{2(\alpha^2 - 1)}$$

■ vertex $2\varphi(1 \leq \varphi \leq n)$ of $P_n$;

$$f(2\varphi, n) = \frac{1}{2} \left[ (2+3)\{ \sum_{\vartheta=1}^{\varphi} \alpha^{2\vartheta-1} + \sum_{\vartheta=1}^{n-\varphi} \alpha^{2\vartheta+1} + 2(\alpha^n + \alpha^{n+1}) \} + (3+3)\{ \alpha^2 + 2 \sum_{\vartheta=1}^{2n-2\varphi-1} \alpha^\vartheta + 2 \sum_{\vartheta=1}^{2n-2\varphi+1} \alpha^{\vartheta+1} \} \right]$$

$$= \frac{1}{2} \left[ (2+3)\frac{\alpha(\alpha^{2\varphi} - 1)}{\alpha^2 - 1} + \frac{\alpha^{3-2\phi}(\alpha^{2n} - \alpha^{2\phi})}{\alpha^2 - 1} + 2(\alpha^n + \alpha^{n+1}) \} + (3+3)\alpha^2 - \frac{2\alpha^{1-2\varphi}(\alpha^{2\varphi} + \alpha^{2n+1})}{\alpha - 1} - \frac{2\alpha^{1-2\varphi}(\alpha^{1-2\varphi} + \alpha^{2n})}{\alpha - 1} \} \right]$$

$$= \frac{1}{2} \Big[ 6\alpha\left( \frac{2\alpha^{2n-2\varphi+1} + 2\alpha^{2n-2\varphi} + \alpha^2 - 3\alpha - 2}{\alpha - 1} \right)$$
$$+ 5\left( 2\alpha^n + 2\alpha^{n+1} + \frac{\alpha(\alpha^{2\varphi} - 1)}{\alpha^2 - 1} + \frac{\alpha^3(\alpha^{2n-2\varphi} - 1)}{\alpha^2 - 1} \right) \Big]$$

■ vertex 2φ+1(1≤φ≤n) of $P_n$;

$$f(2\varphi + 1, n) = \frac{1}{2} \left[ \begin{array}{l} (2+3)(\sum_{\vartheta=1}^{\omega-\varphi} \alpha^{2\vartheta} + \sum_{\vartheta=1}^{\varphi} \alpha^{2\vartheta} + \alpha^{n-1} + \alpha^n + \alpha^{n+1} + \alpha^{n+2}) + \\ +(3+3)(\sum_{\vartheta=1}^{2n-2\varphi-1} \alpha^{\vartheta} + \sum_{\vartheta=1}^{2\varphi} \alpha^{\vartheta} + \sum_{\vartheta=1}^{\varphi-1} \alpha^{2\vartheta} + \sum_{\vartheta=1}^{\varphi-1} \alpha^{2\vartheta+1} + \alpha^{n+1}) \end{array} \right]$$

$$= \frac{1}{2} \left[ 5\left( \frac{\alpha^{2-2\varphi}(\alpha^{2\varphi} - \alpha^{2n})}{\alpha^2 - 1} + \frac{\alpha^2(\alpha^{2\varphi} - 1)}{\alpha^2 - 1} + \alpha^{n-1} + \alpha^n + \alpha^{n+1} + \alpha^{n+2} \right) + 6\left( -\frac{2\alpha^{-2\varphi}(\alpha^{2\varphi+1} - \alpha^{2n})}{\alpha - 1} + \frac{\alpha(\alpha^{2\varphi} - 1)}{\alpha - 1} + \frac{\alpha^{2\varphi} - \alpha^2}{\alpha^2 - 1} - \frac{\alpha(\alpha^2 - \alpha^{2\varphi})}{\alpha^2 - 1} + \alpha^{n+1} \right) \right]$$

$$= \frac{1}{2(\alpha^2 + 1)} (6\alpha^3 + 34\alpha^2 + 18\alpha - 11\alpha^{2\varphi+2} - 12\alpha^{2\varphi+1} - 5\alpha^{n+4} - 11\alpha^{n+3} + 6\alpha^{n+1}$$

$$+5\alpha^n + 5\alpha^{n-1} - 12\alpha^{2n-2\varphi+1} - 12\alpha^{2n-2\varphi} - 5\alpha^{2n-2\varphi+2} - 6\alpha^{2\varphi})$$

vertex $\bar{\varphi}(1 \le \bar{\varphi} \le n)$ of $P_\omega$;

$$f(\bar{\varphi}, n) = \frac{1}{2} \left[ (2+2)\{ \sum_{\vartheta=1}^{n-\varphi+1} \alpha^{2\vartheta} + 2\alpha^{2n} + \alpha^2 \} + 2.(2+2)\{ \alpha + \sum_{\vartheta=1}^{2n-2\varphi} \alpha^{\vartheta+1} \} \right]$$

$$= \frac{1}{2} [4\left( 2\alpha^{2n} + \alpha^2 - \frac{\alpha^{2-2\varphi}(\alpha^{2\varphi} - \alpha^{2n+2})}{\alpha^2 - 1} \right) + 8\left( \alpha + \frac{\alpha^{2-2\varphi}(\alpha^{2n} - \alpha^{2\varphi})}{\alpha - 1} \right)$$

$$= \frac{2\alpha^4 - 9\alpha^2 - 5\alpha + 4\alpha^{2n+2} - 4\alpha^{2n} + 2\alpha^{2n-2\varphi+4} + 5\alpha^{2n-2\varphi+3} + 5\alpha^{2n-2\varphi+2}}{\alpha^2 - 1}$$

According to four vertex of pentagonal chain add the sum of all and divided by two,

$$S_c(P_n) = \frac{1}{2\alpha(\alpha^2 - 1)^2} [\alpha^n(\alpha^2 - 1)\{-5 - \alpha(5 + \alpha(5\alpha + 11)) + n(5 + \alpha(15 +$$

$$\alpha(21 + 5\alpha)))\} + \alpha^2\{-12 + \alpha(-53 + \alpha(-11 + 2\alpha)(5 + 2\alpha)) +$$

$$n(\alpha - 1)(\alpha + 1)(-40 + \alpha(-73 + \alpha(-23 + 8\alpha)))\} + \alpha^{2n+1}\{16 +$$

$$4n(\alpha^2 - 1)^2 + \alpha(24 + \alpha(21 + \alpha(31 + 4\alpha(7 + 2\alpha))))\}]$$

Hence proved.

## 3.6. Theorem

Let $P_n$ be a pentagonal chain graph. Then

$$S_c^*(P_n, \alpha) = \frac{1}{2} \sum_{x,y \subseteq V(P_n)} dx.dy.\alpha^{d(x,y)}$$

**Proof:** Similar to the above proofs, we have

■ vertex 1 of $P_n$;

$$f(n) = \frac{1}{2} \left[ (2.3)(\sum_{\vartheta=1}^{2n-1} \alpha^{\vartheta} + \sum_{\vartheta=1}^{2n-1} \alpha^{\vartheta+1}) + (2.2)(\alpha + \alpha^{2n} + \alpha^{2n+1} + \sum_{\vartheta=1}^{n} \alpha^{2\vartheta}) \right]$$

$$f(n) = \frac{1}{2}\left[(2.3)\left(\frac{\alpha^{2n}-\alpha}{\alpha-1} + \frac{\alpha(\alpha^{2n}-\alpha)}{\alpha-1}\right) + (2.2)\left(\alpha + \alpha^{2n} + \alpha^{2n+1} + \frac{\alpha^2(\alpha^{2n}-1)}{\alpha^2-1}\right)\right]$$

$$= \frac{2\alpha^{2n+3} + 7\alpha^{2n+2} + 4\alpha^{2n+1} + \alpha^{2n} - \alpha^3 - 8\alpha^2 - 5\alpha}{\alpha^2-1}$$

■ vertex $2\varphi(1 \leq \varphi \leq n)$ of $P_\omega$;

$$f(2\varphi, n) = \frac{1}{2}\left[\begin{array}{c}(2.3)\{\sum_{\vartheta=1}^{\varphi}\alpha^{2\vartheta-1} + \sum_{\vartheta=1}^{n-\varphi}\alpha^{2\vartheta+1} + 2(\alpha^{2n}+\alpha^{n+1})\} \\ +(3.3)\{2\sum_{\vartheta=1}^{2n-2\varphi+1}\alpha^{\vartheta} + 2\sum_{\vartheta=1}^{2n-2\varphi-1}\alpha^{\vartheta+1} + \alpha^2\}\end{array}\right]$$

$$= \frac{1}{2}[(2.3)\frac{\alpha(\alpha^{2\varphi}-1)}{\alpha^2-1} + \frac{\alpha^{2-3\varphi}(\alpha^{2n}-\alpha^{2\varphi})}{\alpha^2-1} + 2(\alpha^n + \alpha^{n+1})\} + (3.3)\alpha^2 - \frac{2\alpha^{1-2\varphi}(\alpha^{2\varphi}-\alpha^{1+2n})}{\alpha-1}$$
$$- \frac{2\alpha^{1-2\varphi}(\alpha^{1+2\varphi}-\alpha^{2n})}{\alpha-1}\}]$$

$$= \frac{3}{2(\alpha^2-1)}[8\alpha^{2n-2\varphi+3} + 12\alpha^{2n-2\varphi+2} + 6\alpha^{2n-2\varphi+1} + 4\alpha^{n+3} + 4\alpha^{n+2} - 4\alpha^{n+1} - 4\alpha^n$$
$$+ 2\alpha^{2\varphi+1} + 3\alpha^4 - 8\alpha^3 - 15\alpha^2 - 8\alpha]$$

■ vertex $2\varphi+1(1 \leq \varphi \leq n-1)$ of $P_n$;

$$f(2\varphi+1, n) = \frac{1}{2}\left[\begin{array}{c}(2.3)\{\sum_{\vartheta=1}^{n-\varphi}\alpha^{2\vartheta} + \sum_{\vartheta=1}^{\varphi}\alpha^{2\vartheta} + \alpha^{n-1} + \alpha^n + \alpha^{n+1} + \alpha^{n+2}\} \\ +(3.3)\{\alpha^{n+1} + 2\sum_{\vartheta=1}^{2n-2\varphi-1}\alpha^{\vartheta} + \sum_{\vartheta=1}^{2\varphi}\alpha^{\vartheta} + \sum_{\vartheta=1}^{\varphi-1}\alpha^{2\vartheta} + \sum_{\vartheta=1}^{\varphi-1}\alpha^{2\vartheta+1}\}\end{array}\right]$$

$$= \frac{1}{2}[(2.3)\frac{\alpha^{2-2\varphi}(\alpha^{2n}-\alpha^{2\varphi})}{\alpha-1} + \frac{\alpha^2(\alpha^{2\varphi}-1)}{\alpha-1} + \alpha^{n-1} + \alpha^n + \alpha^{n+1} + \alpha^{n+2}\} + (3.3)\alpha^{n+1}$$
$$+ \frac{2\alpha^{-2\varphi}(\alpha^{2n}-\alpha^{2\varphi+1})}{\alpha-1} + \frac{\alpha(\alpha^{2\varphi}-1)}{\alpha-1} + \frac{\alpha^{2\varphi}-\alpha^2}{\alpha^2-1} + \frac{\alpha(\alpha^{2\varphi}-\alpha^2)}{\alpha^2-1}\}$$

$$= \frac{1}{2}[2\alpha^{2n-2\varphi+2} + 6\alpha^{2n-2\varphi+1} + 6\alpha^{2n-2\varphi} + 2\alpha^{n+4} + 5\alpha^{n+3} - 3\alpha^{n+1} - 2\alpha^n - 2\alpha^{n-1}$$
$$+ 5\alpha^{2\varphi+2} + 6\alpha^{2\varphi+1} + 3\alpha^{2\varphi} - 3\alpha^3 - 16\alpha^2 - 9\alpha]$$

vertex $\bar\varphi (1 \leq \bar\varphi \leq n)$ of $P_n$;

$$f(\bar\varphi, n) = \frac{1}{2}\left[(2.2)\{\alpha^2 + 2\alpha^{2n} + \sum_{\vartheta=1}^{n-\varphi+1}\alpha^{2\vartheta}\} + (2.2.3)\{\alpha + \sum_{\vartheta=1}^{2n-2\varphi}\alpha^{\vartheta+1}\}\right]$$

$$= \frac{1}{2}\left[(2.2)\left\{\alpha^2 + 2\alpha^{2n} + \frac{\alpha^{2-2\varphi}(\alpha^{2n+2}-\alpha^{2\varphi})}{\alpha^2-1}\right\} + (2.2.3)\left\{\alpha + \frac{\alpha^{2-2\varphi}(\alpha^{2n}-\alpha^{2\varphi})}{\alpha-1}\right\}\right]$$

$$= \frac{2\alpha^2(\alpha^2 - 3\alpha - 5)}{\alpha^2 - 1} + \frac{2\alpha^{2n-2\varphi+4}}{\alpha^2 - 1} + \frac{6\alpha^{2n-2\varphi+2}}{\alpha - 1} + 6\alpha + 4\alpha^{2n}$$

Sum of all and divided by two,

$$S_c^*(P_n, \alpha) = \frac{4f(n) + 2\sum_{\varphi=1}^{n} f(2\varphi, n) + 2\sum_{\varphi=1}^{n-1} f(2\varphi+1, n) + \sum_{\varphi=1}^{n} f(\bar{\varphi}, n)}{2}$$

$$S_c^*(P_n, \alpha) = \frac{1}{2\alpha(\alpha^2 - 1)^2}[8\alpha^{2n+6} + \alpha^{2n+5}(4n + 30) + 44\alpha^{2n+4} - \alpha^{2n+3}(8n - 39)$$

$$-38\alpha^{2n+2} + \alpha^{2n+1}(4n + 23) + 6\alpha^{n+7}(n - 1) + 3\alpha^{n+6}(9n - 5) +$$

$$+6\alpha^{n+5}(n + 1) - \alpha^{n+4}(48n - 24) - \alpha^{n+3}(30n - 6) + 3\alpha^{n+2}(5n - 1)$$

$$+6\alpha^{n+1}(3n - 1) + 6\alpha^n(n - 1) + 11\alpha^7 - \alpha^6(33n - 5) - \alpha^5(14n + 7)$$

$$-2\alpha^4(12n + 35) + \alpha^3(103n - 85) + \alpha^2(57n - 25)]$$

Hence proved.

## 4. The comparison of TIs

The computation of considered indices provides valuable insights into the structural characteristics of the pentagonal chain graph. The comparative values of certain distance-based TIs are shown in Table 1. The Wiener index offered information about the overall connectivity and efficiency of the graph, while the Gutman index provided insights into the presence of conjugated circuits. The Schultz index and modified Schultz index shed light on the dispersion of distances within the graph, indicating its potential for different applications. We see as graphically representation of all indices in Fig 2 and it is easy to describe that the modified Schultz polynomial index give better result for the physically and chemical property of molecules as compare to the other considered TI.

Table 1. Comparative values of some distance-based TIs.

| n | WI | GI | SI | M.SI | SP | M.SP |
|---|------|--------|---------|-----------|----------------|----------------|
|   | $W(P_n)$ | $Gut(P_n)$ | $S_c(P_n)$ | $S_c^*(P_n)$ | $S_c(P_n, \alpha)$ | $S_c^*(P_n, \alpha)$ |
| 1 | 36 | 182 | 81 | 91 | 170 | 190 |
| 2 | 159 | 939 | 387 | 855/2 | 1077 | 1273 |
| 3 | 420 | 2664 | 1059 | 1206 | 4796 | 5738 |
| 4 | 869 | 5749 | 2237 | 5295/2 | 19514 | 23353 |
| 5 | 1556 | 10586 | 4061 | 4873 | 77988 | 92992 |

## 5. Conclusion

This study gives the importance of distance-based topological indices in understanding the properties of a pentagonal chain network. These indices provide useful insights into networks' physicochemical properties and structural characterizations. Since, the QSAR correlate the structure of any networks with their biological activity or other properties. Distance-based topological indices provide an important descriptor in QSAR models, to assists the prediction

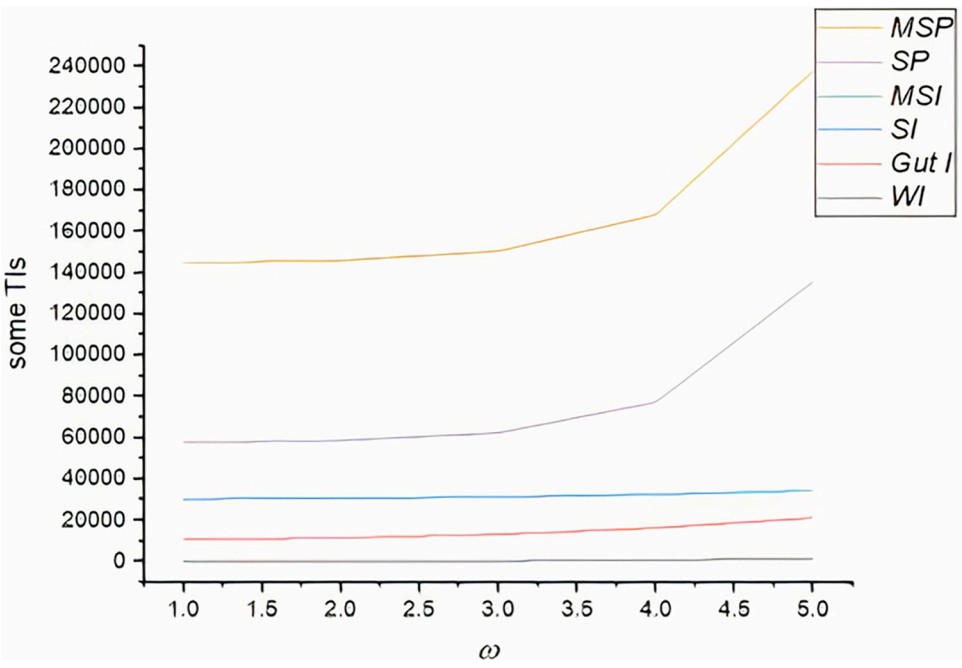

**Fig 2. Comparative values of some distance-based TIs.**

of molecular activity and guide the design of novel compounds with optimized properties. In the near future we aim to calculate the resistance distance based topological indices for the pentagonal chain network.

## Acknowledgments

The authors are grateful to the reviewers for their valuable suggestions.

## Author Contributions

**Conceptualization:** Guofeng Yu, Shahid Zaman, Mah Jabeen, Xuewu Zuo.

**Data curation:** Guofeng Yu, Shahid Zaman, Xuewu Zuo.

**Formal analysis:** Guofeng Yu, Shahid Zaman, Mah Jabeen, Xuewu Zuo.

**Funding acquisition:** Guofeng Yu, Xuewu Zuo.

**Investigation:** Guofeng Yu, Shahid Zaman, Mah Jabeen, Xuewu Zuo.

**Methodology:** Guofeng Yu, Shahid Zaman, Mah Jabeen, Xuewu Zuo.

**Project administration:** Guofeng Yu.

**Resources:** Guofeng Yu, Shahid Zaman, Xuewu Zuo.

**Software:** Guofeng Yu, Shahid Zaman, Xuewu Zuo.

**Supervision:** Guofeng Yu, Xuewu Zuo.

**Validation:** Guofeng Yu, Shahid Zaman, Mah Jabeen, Xuewu Zuo.

**Visualization:** Guofeng Yu, Shahid Zaman, Xuewu Zuo.

**Writing – original draft:** Guofeng Yu, Shahid Zaman, Mah Jabeen, Xuewu Zuo.

**Writing – review & editing:** Guofeng Yu, Shahid Zaman, Mah Jabeen, Xuewu Zuo.

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
