## [Decision Letter · Decision Letter 0]

13 Mar 2024

PONE-D-23-40415The study of pentagonal chain with respect to schultz index, modified schultz index, schultz polynomial and modified schultz polynomialPLOS ONE

Dear Dr. Zaman,

Thank you for submitting your manuscript to PLOS ONE. After careful consideration, we feel that it has merit but does not fully meet PLOS ONE’s publication criteria as it currently stands. Therefore, we invite you to submit a revised version of the manuscript that addresses all the points raised during the review process. Please submit your revised manuscript by Apr 27 2024 11:59PM. If you will need more time than this to complete your revisions, please reply to this message or contact the journal office at plosone@plos.org. Please include the following items when submitting your revised manuscript:A rebuttal letter that responds to each point raised by the academic editor and reviewer(s). You should upload this letter as a separate file labeled 'Response to Reviewers'.A marked-up copy of your manuscript that highlights changes made to the original version. You should upload this as a separate file labeled 'Revised Manuscript with Track Changes'.An unmarked version of your revised paper without tracked changes. You should upload this as a separate file labeled 'Manuscript'.

We look forward to receiving your revised manuscript.

Kind regards,

Claudio Zandron

Academic Editor

PLOS ONE

Journal Requirements: 

Reviewers' comments:

Reviewer's Responses to Questions

**Comments to the Author**

1. Is the manuscript technically sound, and do the data support the conclusions?

Reviewer #1: Yes

Reviewer #2: Yes

Reviewer #3: Yes

2. Has the statistical analysis been performed appropriately and rigorously? 

Reviewer #1: Yes

Reviewer #2: Yes

Reviewer #3: N/A

3. Have the authors made all data underlying the findings in their manuscript fully available?

Reviewer #1: No

Reviewer #2: Yes

Reviewer #3: Yes

4. Is the manuscript presented in an intelligible fashion and written in standard English?

Reviewer #1: Yes

Reviewer #2: Yes

Reviewer #3: No

5. Review Comments to the Author

Reviewer #1: Kindly discuss the real time applications on several significant distance-based topological indices for pentagonal

chain graphs including the imperial index (wiener index), Gutman index, Schultz index,

modified Schultz index, Schultz polynomial, and modified Schultz polynomials on Particular example.

Reviewer #2: In this paper, the authors studied pentagonal chain with respect to schultz index, modified schultz index, schultz polynomial and modified schultz polynomial. The significance of this research is in structural properties and relationships between atoms in molecules. These indices have found applications in various areas of chemistry, including drug design, material science, and chemical information analysis. It also underscores the practical implications of these findings in quantitative structure-property and structure-activity relationships, which have implications for drug development and chemical analysis. Based on above, it seems that this article has some merits to be considered for publication in PLOS ONE. However, the following Major revisions must be addressed before formal acceptance. My point-wise comments are as follows.

1. The abstract of this paper is not attractive. Please make it to the point.

2. Introduction section: This section must be further enhanced by including and discussing some more latest relevant studies. Also, the authors should add some literature about “pentagonal chain”.

3. In the 1st line of introduction, it seems that the authors have denote a graph by “G=(V_G,E_G)” so in the whole manuscript they must use the notation of vertex set and edge set according to this.

4. In the second paragraph of introduction, the authors have used some notations without defined them. Similar, problem appears in many places of this paper. Please define all the notations and terminologies at first and then use them.

5. Page#4: Please revise the sentence: “Their indices' Schultz polynomial.”

6. Please revise the statement of Theorem 4.1.

7. In the whole manuscript what is “ ” ?

8. In the Proof of theorem 4.1, the authors have skip some important mathematical steps, please include them.

9. In the proof of theorem 4.1: “Point 1 of ” should be revise as “vertex 1 of :” Similar, problem repeats in the remaining proof of the theorems.

10. I noticed that the original code/data for this article is not available. If in case the authors have developed codes for computations, it is suggested to make it open-source, its popularity can be further increased. There is no need for it if the computations are done manually.

11. It will further enhance the quality of this work if the current results are compared with some more latest relevant studies.

12. In the results section, add one or more paragraphs about the physical significance of their results in order to prove the importance and application their findings.

13. Conclusion needs further improvement and future research directions should be added.

Reviewer #3: Use standard notation. points should be replace by vertices, don't use Greek words to represent vertices for example replace omega by n and so on. paper is not well written. Give details of proof. At the moment paper has some formulas that can hard to verify for larger values.

In Table 1, some remarks were written about the finding. Explain

Conclusion section need to improve, the authors need to justify claim "These metrics are

employed to forecast the chemical and physical characteristics of molecules"

6. PLOS authors have the option to publish the peer review history of their article (what does this mean?). If published, this will include your full peer review and any attached files.

Reviewer #1: No

Reviewer #2: No

Reviewer #3: No

---

## [Author Response · Author response to Decision Letter 0]

21 Mar 2024

Response/compliance to referee comments/suggestions

The editor,

PLOS ONE.

Subject: Submission of revised manuscript 

Respected Editor,

We would like to express our sincere thanks to you and the reviewers for valuable comments to improve our manuscript with title “The study of pentagonal chain with respect to schultz index, modified schultz index, schultz polynomial and modified schultz polynomial”.

We have read the comments carefully and revised the manuscript by following the reviewers' directions. The revised parts of the manuscript are colored yellow.

The detailed responses to reviewers’ comments are given below:

Referee 1:

Comment 1. Kindly discuss the real time applications on several significant distance-based topological indices for pentagonal chain graphs including the imperial index (wiener index), Gutman index, Schultz index,

modified Schultz index, Schultz polynomial, and modified Schultz polynomials on Particular example.

Response/compliance: Thank you very much for your invaluable suggestions. We have added the applications of the considered topological indices in the abstract and introduction. Meanwhile, we have cited some interesting articles which support to the real time applications of our main results.

Referee 2:

Comment 1). The abstract of this paper is not attractive. Please make it to the point.

Response/compliance: Thank you for your comment. We have revised the abstract by adding some interesting applications and focused only our contribution. 

Comment 2). Introduction section: This section must be further enhanced by including and discussing some more latest relevant studies. Also, the authors should add some literature about “pentagonal chain”.

Response/compliance: We have done accordingly.

Comment 3). In the 1st line of introduction, it seems that the authors have denote a graph by “G=(V_G,E_G)” so in the whole manuscript they must use the notation of vertex set and edge set according to this.

Response/compliance: We have done accordingly.

Comment 4). In the second paragraph of introduction, the authors have used some notations without defined them. Similar, problem appears in many places of this paper. Please define all the notations and terminologies at first and then use them.

Response/compliance: We have done accordingly.

Comment 5). Page#4: Please revise the sentence: “Their indices' Schultz polynomial.”

Response/compliance: We have revised accordingly.

Comment 6). In the whole manuscript what is “ ” ?.

Response/compliance: We have defined this term.

Comment 7). In the Proof of theorem 4.1, the authors have skip some important mathematical steps, please include them.

Response/compliance: We have added those steps.

Comment 8). In the proof of theorem 4.1: “Point 1 of ” should be revise as “vertex 1 of :” Similar, problem repeats in the remaining proof of the theorems.

Response/compliance: We have revised accordingly.

Comment 9). I noticed that the original code/data for this article is not available. If in case the authors have developed codes for computations, it is suggested to make it open-source, its popularity can be further increased. There is no need for it if the computations are done manually.

Response/compliance: Thank you for your comment. Since, our results only based on mathematical calculations, we did not use any code/data.

Comment 10). It will further enhance the quality of this work if the current results are compared with some more latest relevant studies.

Response/compliance: We have done accordingly.

Comment 11). In the results section, add one or more paragraphs about the physical significance of their results in order to prove the importance and application their findings.

Response/compliance: We have done accordingly.

Comment 12). Conclusion needs further improvement and future research directions should be added.

Response/compliance: We have done accordingly.

Referee 3:

Use standard notation. points should be replace by vertices, don't use Greek words to represent vertices for example replace omega by n and so on. paper is not well written. Give details of proof. At the moment paper has some formulas that can hard to verify for larger values. In Table 1, some remarks were written about the finding. Explain Conclusion section need to improve, the authors need to justify claim "These metrics are employed to forecast the chemical and physical characteristics of molecules"

Response/compliance: Thank you very much for your invaluable suggestions. In the updated version we have replaced the word “points” with vertices, “Omega” with “n” also replaced various notations with the standard notations. We have added some more important calculations in the proofs of our theorems which will be helpful to the readers for better understanding of these mathematical results. Indeed, the formulas are for general pentagons, but it is easy to verify for one pentagon, two pentagons, and up-to so-on, as given in Table 1.

We hope that these revisions are satisfactory and that the revised version will be acceptable for publication in PLOS ONE. Thank you very much

Yours sincerely

Dr. Shahid Zaman

---

## [Decision Letter · Decision Letter 1]

7 May 2024

PONE-D-23-40415R1The study of pentagonal chain with respect to schultz index, modified schultz index, schultz polynomial and modified schultz polynomialPLOS ONE

Dear Dr. Zaman,

Thank you for submitting your manuscript to PLOS ONE. After careful consideration, we feel that it has merit but does not fully meet PLOS ONE’s publication criteria as it currently stands. Therefore, we invite you to submit a revised version of the manuscript that addresses the points raised during the review process.

We look forward to receiving your revised manuscript.

Kind regards,

Claudio Zandron

Academic Editor

PLOS ONE

Journal Requirements:

Additional Editor Comments:

One reviewer highlights some minor corrections to be done before the paper can be published. Please consider all comments and correct the paper accordingly.

Reviewers' comments:

Reviewer's Responses to Questions

**Comments to the Author**

1. If the authors have adequately addressed your comments raised in a previous round of review and you feel that this manuscript is now acceptable for publication, you may indicate that here to bypass the “Comments to the Author” section, enter your conflict of interest statement in the “Confidential to Editor” section, and submit your "Accept" recommendation.

Reviewer #1: All comments have been addressed

Reviewer #2: (No Response)

2. Is the manuscript technically sound, and do the data support the conclusions?

Reviewer #1: Yes

Reviewer #2: Yes

3. Has the statistical analysis been performed appropriately and rigorously? 

Reviewer #1: Yes

Reviewer #2: N/A

4. Have the authors made all data underlying the findings in their manuscript fully available?

Reviewer #1: Yes

Reviewer #2: Yes

5. Is the manuscript presented in an intelligible fashion and written in standard English?

Reviewer #1: Yes

Reviewer #2: Yes

6. Review Comments to the Author

Reviewer #1: The Authors have addressed the comments raised in previous review. The manuscript is presented in standard way.

Reviewer #2: The authors have incorporated my previous suggestion. However, there are still some errors\\typos exist in the paper. I invite the authors to make them correct. Based on the below points I recommend a Minor revisions. My point-wise comments are as follows.

1. In abstract “It also uses to predict the…” should be revised as “ It also predict the… ”

2. In abstract: “we consider a pentagonal…” should be revised as “ we have considered a pentagonal…”

3. Please add one more keyword.

4. The heading of section 3 should be revised.

5. The statement of Theorem 4.3 is not appropriate.

6. Please explain table 1 further. Also explain the comparison graphs in Fig.2.

7. Add some physio chemical applications in the conclusion section.

7. PLOS authors have the option to publish the peer review history of their article (what does this mean?). If published, this will include your full peer review and any attached files.

Reviewer #1: No

Reviewer #2: No

---

## [Author Response · Author response to Decision Letter 1]

14 May 2024

Response/compliance to referee comments/suggestions

The editor,

PLOS ONE.

Subject: Submission of the 2nd revised manuscript

Respected Editor,

We would like to express our sincere thanks to you and the reviewers for valuable comments to improve our manuscript with title “The study of pentagonal chain with respect to schultz index, modified schultz index, schultz polynomial and modified schultz polynomial”.

We have read the comments carefully and revised the manuscript by following the reviewers' directions. The revised parts of the manuscript are colored yellow.

The detailed responses to reviewers’ comments are given below:

Referee 2:

Comment 1. In abstract “It also uses to predict the…” should be revised as “ It also predict the… ”

Response/compliance: Thank you very much for your invaluable suggestions. We have revised accordingly.

Comment 2). In abstract: “we consider a pentagonal…” should be revised as “ we have considered a pentagonal…”

Response/compliance: Thank you for your comment. We have revised accordingly.

Comment 3). Please add one more keyword.

Response/compliance: We have done accordingly.

Comment 4). The heading of section 3 should be revised.

Response/compliance: We have done accordingly.

Comment 5). The statement of Theorem 4.3 is not appropriate.

Response/compliance: We have revised it.

Comment 6). Add some physio chemical applications in the conclusion section.

Response/compliance: We have added in the revision.

We hope that these revisions are satisfactory and that the revised version will be acceptable for publication in PLOS ONE. Thank you very much

Yours sincerely: Dr. Shahid Zaman

---

## [Editor Report · Decision Letter 2]

17 May 2024

The study of pentagonal chain with respect to schultz index, modified schultz index, schultz polynomial and modified schultz polynomial

PONE-D-23-40415R2

Dear Dr. Zaman,

We’re pleased to inform you that your manuscript has been judged scientifically suitable for publication and will be formally accepted for publication once it meets all outstanding technical requirements.

Kind regards,

Claudio Zandron

Academic Editor

PLOS ONE
---

## [Editor Report · Acceptance letter]

21 May 2024

PONE-D-23-40415R2 

PLOS ONE

Dear Dr. Zaman, 

I'm pleased to inform you that your manuscript has been deemed suitable for publication in PLOS ONE. Congratulations! Your manuscript is now being handed over to our production team.

Kind regards, 

on behalf of

Dr. Claudio Zandron 

Academic Editor

PLOS ONE